# Risk factors for mechanical ventilation and ECMO in COVID-19 patients admitted to the ICU: A multicenter retrospective observational study

Ryo Takada[1], Tomonori Takazawa[1]*, Yoshihiko Takahashi[2], Kenji Fujizuka[2], Kazuki Akieda[3], Shigeru Saito[4]

1 Intensive Care Unit, Gunma University Hospital, Maebashi, Gunma, Japan, 2 Advanced Medical Emergency Department and Critical Care Center, Japan Red Cross Maebashi Hospital, Maebashi, Gunma, Japan, 3 Department of Emergency Medicine, Subaru Health Insurance Society Ota Memorial Hospital, Ota, Gunma, Japan, 4 Department of Anesthesiology, Gunma University Graduate School of Medicine, Maebashi, Gunma, Japan

* takazawt@gunma-u.ac.jp

## Abstract

### Background

The primary purpose of this study was to investigate risk factors associated with the need for mechanical ventilation (MV) and extracorporeal membrane oxygenation (ECMO) in COVID-19 patients admitted to the intensive care unit (ICU).

### Methods

We retrospectively enrolled 66 consecutive COVID-19 patients admitted to the ICUs of three Japanese institutions from February 2020 to January 2021. We performed logistic regression analyses to identify risk factors associated with subsequent MV and ECMO requirements. Further, multivariate analyses were performed following adjustment for Acute Physiology and Chronic Health Evaluation (APACHE) II scores.

### Results

At ICU admission, the risk factors for subsequent MV identified were: higher age (Odds Ratio (OR) 1.04, 95% Confidence Interval (CI) 1.00–1.08, P = 0.03), higher values of APACHE II score (OR 1.20, 95% CI 1.08–1.33, P < 0.001), Sequential Organ Failure Assessment score (OR 1.53, 95% CI 1.18–1.97, P < 0.001), lactate dehydrogenase (LDH) (OR 1.01, 95% CI 1.00–1.02, p<0.001) and C-reactive protein (OR 1.09, 95% CI 1.00–1.19, P = 0.04), and lower values of lymphocytes (OR 1.00, 95% CI 1.00–1.00, P = 0.02) and anti-thrombin (OR 0.95, 95% CI 0.91–0.95, P < 0.01). Patients who subsequently required ECMO showed lower values of estimated glomerular filtration rate (OR 0.98, 95% CI 0.96–1.00, P = 0.04) and antithrombin (OR 0.94, 95% CI 0.88–1.00, P = 0.03) at ICU admission. Multivariate analysis showed that higher body mass index (OR 1.19, 95% CI 1.00–1.40, P = 0.04) and higher levels of LDH (OR 1.01, 95% CI 1.01–1.02, P < 0.01) were independent

**Data Availability Statement:** All relevant data are within the paper and its Supporting Information files.

**Funding:** The authors received no specific funding for this work.

**Competing interests:** The authors have declared that no competing interests exist.

**Abbreviations:** ARDS, Acute respiratory distress syndrome; ANOVA, Analysis of variance; APACHE II score, Acute physiology and chronic health evaluation II score; CI, Confidence interval; COVID-19, Coronavirus disease 2019; DIC, Disseminated intravascular coagulopathy; ECMO, Extra-corporeal membrane oxygenation; eGFR, estimated glomerular filtration rate; ICU, Intensive care unit; OR, Odds ratio; SARS-CoV-2, severe acute respiratory syndrome coronavirus 2; SOFA score, Sequential organ failure assessment score.

risk factors for the need for MV. Lower level of antithrombin (OR 0.94, 95% CI 0.88–1.00, P = 0.03) was a risk factor for the need for ECMO.

## Conclusion

We showed that low antithrombin level at ICU admission might be a risk factor for subsequent ECMO requirements, in addition to other previously reported factors.

## Introduction

Currently, as of September 2022, although the number of severe cases has diminished as compared to the time of the delta and earlier variants, the global COVID-19 pandemic caused by severe acute respiratory syndrome coronavirus 2 (SARS-CoV-2) is still ongoing. A small number of patients still become critically ill and require mechanical ventilation (MV), and some of them require temporary lung rest with extracorporeal circulation. As during the delta and earlier waves, a potential future increase in the number of severe cases would likely again lead to a shortage of intensive care unit (ICU) beds and various medical resources, such as mechanical ventilators, extracorporeal circulators, and skilled medical staff [1].

Since the early stages of the pandemic, many retrospective studies have been published on patient risk factors predicting COVID-19 mortality. For example, Sequential Organ Failure Assessment (SOFA) score [2, 3], advanced age [2–8], low lymphocyte count [5, 8, 9], and high lactate dehydrogenase (LDH) [5, 8] and d-dimer [2] levels have been repeatedly identified as risk factors for severe disease and death.

The course of COVID-19 in individual patients is variable, and there are several cases of hypoxemia requiring MV within a few days of symptom onset. Patients who require MV or extracorporeal membrane oxygenation (ECMO) tend to have a prolonged treatment period, resulting in a worse prognosis and increased consumption of medical resources [10–12]. Although many studies have been published to predict the mortality of severe COVID-19 patients, few have investigated the association between patient risk factors at ICU admission and the subsequent requirements of MV or ECMO. Prediction of COVID-19 patients subsequently requiring MV or ECMO might help prevent the disease's deterioration through early intervention. This study aimed to explore patient risk factors that are likely to predispose them to the need for MV and ECMO by analyzing the data of COVID-19 patients admitted to the ICU.

## Methods

### Study design

We conducted a retrospective observational study of severe COVID-19 patients. The study included all consecutive patients admitted to the ICUs of three hospitals in Gunma Prefecture, Japan: Gunma University Hospital, Japanese Red Cross Maebashi Hospital, and Subaru Health Insurance Society Ota Memorial Hospital. These hospitals treat severe COVID-19 patients and are equipped with mechanical ventilators and ECMO devices. The case enrollment period was from February 9, 2020, to January 31, 2021, before the alpha strain epidemic in Japan. The patients were followed up for at least three months after admission to the ICU. The diagnosis of COVID-19 was made by a polymerase chain reaction (PCR) test or a quantitative antigen test. Eligibility criteria were patients who received intensive care for COVID-19 at any of the

three hospitals participating in the study within the study period. Exclusion criteria were patients who were deemed unsuitable for study inclusion by the physician for ethical reasons, and patients who did not wish to be included in the study after reading the disclosure notice. The decision to introduce MV and ECMO was at the discretion of the attending physicians. Although we have no standardized protocol for the introduction of MV and ECMO, published standard guidelines were used as references [13, 14].

### Ethics approval and informed consent

The ethics committee of each participating hospital (Gunma University Hospital: HS2020-197, Japanese Red Cross Maebashi Hospital: 2021–6, and Subaru Health Insurance Society Ota Memorial Hospital: OR20048) approved this study, and it was performed according to the guidelines of the Declaration of Helsinki. Since the ethics committees of the participating institutions waived the need for obtaining verbal or written consent from the patients participating in this study, patient informed consent was obtained in the form of the option to opt-out on the institutions' websites.

### Variables

The primary endpoint of this study was the impact of patient factors at ICU admission on subsequent MV and ECMO requirements. Possible patient risk factors at admission that might have affected the outcomes, including patient characteristics, preexisting diseases and blood test results, were collected and analyzed. The APACHE II score [15] and SOFA score [16], which are indices of patient severity, were calculated based on data at ICU admission. The preexisting diseases and blood tests for analysis were selected based on similar previous studies.

We also collected data on medications, renal replacement therapy and airway management during ICU admission that might have affected patient outcomes, although they were not likely to be directly related to the primary endpoints.

### Statistical methods

We performed logistic regression analysis to compare the two patient groups, those who required MV and those who did not, and odds ratios (ORs) and 95% confidence intervals were calculated. Similar comparisons were performed between the MV-only group and ECMO group. Kruskal-Wallis ANOVA with the post-hoc Bonferroni test was used to compare patient outcomes between groups. P-values were two-tailed, and those less than 0.05 were considered statistically significant. All statistical analyses were performed with R software (The R Foundation for Statistical Computing, Vienna, Austria).

## Results

Sixty-six consecutive COVID-19 patients admitted to the ICUs of the three hospitals during the study period were included. No patients were excluded from the study. Forty-two of the 66 patients (67%) were treated with MV, and 20 (30.3%) required ECMO. Seven patients died while in the hospital. Table 1 shows baseline patient characteristics at the time of ICU admission.

The covariates associated with the need for MV were as follows: higher age (OR 1.04, 95% CI 1.00–1.08, P = 0.03) and higher levels of APACHE II score (OR 1.20, 95% CI 1.08–1.33, P < 0.001), SOFA score (OR 1.53, 95% CI 1.18–1.97, P < 0.001), C-reactive protein (OR 1.09, 95% CI 1.00–1.19, P = 0.04) and LDH (OR 1.01, 95% CI 1.00–1.02, P < 0.001), and lower levels of antithrombin (OR 0.95, 95% CI 0.91–0.98, P = 0.005) at the time of ICU admission (Table 2). The covariates associated with the need for ECMO were lower levels of antithrombin

**Table 1. Patient baseline characteristics at the time of intensive care unit admission.**

| | All (n = 66) |
|---|---|
| **Demographics** | |
| Age, years | 67 (53–71) |
| Sex, male | 47 (71.2%) |
| Body mass index, kg/m$^2$ | 26.1 (23.4–29.0) |
| **Race** | |
| Asian | 56 (84.8%) |
| Latino | 10 (15.2%) |
| **Scoring at ICU admission** | |
| APACHE II | 15 (9–21) |
| SOFA | 4 (2–7) |
| **Hospital** | |
| Gunma University Hospital | 24 (36%) |
| Japanese Red Cross Maebashi Hospital | 23 (35%) |
| Subaru Health Insurance Society Ota Memorial Hospital | 19 (29%) |
| **Comorbidities** | |
| Cancer | 3 (4.6%) |
| Diabetes mellitus | 29 (43.9%) |
| Obesity | 12 (18.2%) |
| Hypertension | 37 (56.1%) |
| Chronic heart failure | 4 (6.1%) |
| Angina/previous myocardial infarction | 7 (10.6%) |
| Chronic kidney disease | 13 (19.7%) |
| Dialysis | 7 (10.6%) |
| **Laboratory data at admission** | |
| White blood cells, ×10$^9$/mL | 7650 (4950–9925) |
| Lymphocytes, ×10$^6$/mL | 680 (480–960) |
| Platelet count, ×10$^3$/mL | 174.0 (125.3–228.8) |
| Lactate dehydrogenase, U/L | 347 (270–466) |
| Creatine kinase, U/L | 86 (49–170) |
| Aspartate transaminase, U/L | 37 (28–54) |
| Alanine transaminase, U/L | 27 (17–40) |
| Total bilirubin, mg/dL | 0.6 (0.4–0.8) |
| Blood urea nitrogen, mg/dL | 18 (13–32) |
| Serum creatinine, mg/dL | 0.79 (0.62–1.27) |
| eGFR, mL/min/m$^2$ | 70.0(42.0–88.6) |
| C-reactive protein, mg/dL | 7.52 (3.83–13.46) |
| PT-INR | 1.04 (0.99–1.11) |
| APTT, sec | 33.5 (30.2–38.2) |
| Fibrinogen, mg/dL | 546 (452–616) |
| Antithrombin activity †, % | 82.5 (75.1–96.0) |
| D-dimer, µg/mL | 1.5 (1.0–2.4) |
| **Heparin usage before admission** | 5 (7.5%) |

Data are presented as the median (interquartile range) or number (%). APACHE II, Acute Physiology and Chronic Health Evaluation II score; SOFA, Sequential Organ Failure Assessment score; COPD, chronic obstructive pulmonary disease; eGFR, estimated glomerular filtration rate; PT-INR, the international normalized ratio of prothrombin time; APTT, activated partial thromboplastin time. Obesity was defined as a body mass index of over 30 kg/m$^2$. † Four patients in the no-mechanical ventilation group lacked data on antithrombin activity.

**Table 2. Factors correlating with the need for mechanical ventilation in COVID-19 patients.**

| | No mechanical ventilation (n = 24) | Mechanical ventilation (n = 42) | Odds ratio (95% CI) | P-value | Adjusted odds ratio (95% CI) | P-value |
|---|---|---|---|---|---|---|
| Age, years | 60 (46–71) | 69 (55–71) | 1.04 (1.00–1.08) | 0.03 | NA | |
| Sex, male | 17 (68.0%) | 31 (73.8%) | 1.41 (0.43–4.20) | 0.54 | 1.59 (0.44–5.73) | 0.48 |
| Body mass index, kg/m$^2$ | 25.4 (22.8–29.5) | 26.4 (23.5–28.9) | 1.04 (0.93–1.17) | 0.49 | 1.19 (1.00–1.40) | 0.04 |
| APACHE II score | 8 (7–12) | 18 (13–25) | 1.20 (1.08–1.33) | <0.001 | NA | |
| SOFA score | 3 (2–4) | 6 (3–9) | 1.53 (1.18–1.97) | <0.001 | NA | |
| Cancer | 1 (4.0%) | 2 (4.8%) | 1.15 (0.10–13.40) | 0.91 | 1.01 (0.01–16.70) | 0.99 |
| Diabetes mellitus | 9 (36.0%) | 20 (47.6%) | 1.52 (0.54–4.22) | 0.43 | 1.05 (0.32–3.49) | 0.93 |
| Obesity | 6 (24.0%) | 6 (14.3%) | 0.50 (0.14–1.77) | 0.28 | 0.81 (0.18–3.56) | 0.78 |
| Hypertension | 11 (44.0%) | 27 (64.3%) | 2.52 (0.90–7.05) | 0.08 | 1.95 (0.59–6.46) | 0.27 |
| Chronic heart failure | 2 (8.0%) | 2 (4.8%) | 0.55 (0.07–4.18) | 0.56 | 0.20 (0.02–2.16) | 0.18 |
| Angina/previous MI | 4 (16.0%) | 3 (7.1%) | 0.39 (0.08–1.89) | 0.24 | 0.16 (0.02–1.09) | 0.06 |
| Chronic kidney disease | 6 (24.0%) | 7 (16.7%) | 0.60 (0.18–2.05) | 0.42 | 0.34 (0.08–1.56) | 0.17 |
| Dialysis | 3 (12.0%) | 4 (9.5%) | 0.74 (0.15–3.61) | 0.71 | 0.18 (0.02–1.28) | 0.09 |
| White blood cells, ×10$^9$/mL | 6600 (4700–8800) | 8300 (6810–10248) | 1.00 (1.00–1.00) | 0.19 | NA | |
| Lymphocytes, ×10$^6$/mL | 1331 (744–2145) | 630 (430–898) | 1.00 (1.00–1.00) | 0.02 | 1.00 (1.00–1.00) | 0.12 |
| Platelet count, ×10$^3$/mL | 175.0 (132.0–247.0) | 166.0 (125.8–211.3) | 1.00 (1.00–1.00) | 0.34 | 1.00 (1.00–1.00) | 0.41 |
| Lactate dehydrogenase, U/L | 272 (224–339) | 397 (329–514) | 1.01 (1.00–1.02) | <0.001 | 1.01 (1.00–1.02) | <0.01 |
| Creatine kinase, U/L | 53 (40–170) | 100 (61–212) | 1.00 (1.00–1.01) | 0.18 | 1.00 (1.00–1.01) | 0.22 |
| Aspartate transaminase, U/L | 35 (26–44) | 39 (29–60) | 1.02 (1.00–1.05) | 0.12 | 1.02(0.99–1.06) | 0.18 |
| Alanine transaminase, U/L | 33 (16–41) | 27 (17–39) | 1.00 (0.98–1.02) | 0.99 | 1.01(0.98–1.03) | 0.63 |
| Total bilirubin, mg/dL | 0.55 (0.46–0.7) | 0.61 (0.41–0.9) | 3.07 (0.59–16.00) | 0.18 | 3.10 (0.45–21.20) | 0.25 |
| Blood urea nitrogen, mg/dL | 15 (11–22) | 20 (15–33) | 1.02 (0.99–1.00) | 0.21 | 1.00 (0.97–1.03) | 0.83 |
| Serum creatinine, mg/dL | 0.77 (0.64–0.93) | 0.84 (0.62–1.39) | 1.05 (0.90–1.23) | 0.53 | NA | |
| eGFR, mL/min/m$^2$ | 80.9 (61.2–91.2) | 69.4 (40.2–84.0) | 0.99 (0.978–1.01) | 0.25 | NA | |
| C-reactive protein, mg/dL | 4.64 (2.14–11.2) | 9.65 (6.5–14.68) | 1.09 (1.00–1.19) | 0.04 | 1.10 (0.99–1.22) | 0.08 |
| PT-INR | 1.01 (0.95–1.11) | 1.04 (1.0–1.1) | 9.02 (0.08–1040.0) | 0.36 | 2.92 (0.03–273.00) | 0.64 |
| APTT, sec | 33.8 (30.1–36.9) | 33.4 (30.4–38.7) | 1.00 (0.95–1.06) | 0.88 | 1.02 (0.96–1.09) | 0.50 |
| Fibrinogen, mg/dL | 551 (452–616) | 54 7(453–629) | 1.00 (1.00–1.00) | 0.89 | 1.00 (1.00–1.01) | 0.76 |
| Antithrombin activity, % | 92.6 (83.0–101.8) | 77.8 (74.2–88.8) | 0.95 (0.91–0.98) | < 0.01 | 0.97 (0.93–1.01) | 0.13 |
| D-dimer, μg/mL | 1.3 (0.8–1.6) | 1.7 (1.1–2.7) | 1.01 (0.96–1.05) | 0.78 | 1.0 2(0.97–1.06) | 0.51 |

Multivariate analyses were performed after adjusting for disease severity at ICU admission based on APACHE II scores. Data are presented as the median (interquartile range) or number (%). NA, not applicable; APACHE II, Acute Physiology and Chronic Health Evaluation II score; SOFA, Sequential Organ Failure Assessment score; MI, myocardial infarction; eGFR, estimated glomerular filtration rate; PT-INR, the international normalized ratio of prothrombin time; APTT, activated partial thromboplastin time.

(OR 0.94, 95% CI 0.88–1.00, P = 0.03) and estimated glomerular filtration rate (eGFR) (OR 0.98, 95% CI 0.96–1.00, P = 0.04) at the time of ICU admission (Table 3). In addition, we used multivariate logistic regression models adjusted for APACHE II score, because the severity of illness at the time of ICU admission was likely to have been different among the patient groups. After adjustment for APACHE II scores, patients who required MV had higher BMI (OR 1.19, 95% CI 1.00–1.40, P = 0.04) and higher levels of LDH (OR 1.01, 95% CI 1.01–1.02, P < 0.01), while those who received ECMO had lower levels of antithrombin (OR 0.94, 95% CI 0.88–1.00, P = 0.03) at ICU admission (Tables 2 and 3).

**Table 3. Factors correlating with the need for ECMO in COVID-19 patients.**

| | Mechanical ventilation-only (n = 22) | ECMO (n = 20) | Odds ratio (95% CI) | P-value | Adjusted odds ratio (95% CI) | P-value |
|---|---|---|---|---|---|---|
| Age | 69 (60–70) | 67 (53–72) | 0.97 (0.92–1.03) | 0.35 | NA | |
| Sex, Male | 14 (63.6%) | 17 (85.0%) | 3.24 (0.72–14.60) | 0.13 | 3.56 (0.76–16.80) | 0.11 |
| Body mass index, kg/m$^2$ | 25.6 (23.0–28.4) | 26.5 (24.4–29.4) | 1.04 (0.90–1.21) | 0.60 | 1.05 (0.90–1.22) | 0.57 |
| APACHE II score | 16 (13–24) | 20 (13–25) | 1.01 (0.95–1.09) | 0.69 | NA | |
| SOFA score | 5 (3–8) | 7 (4–9) | 1.12 (0.94–1.33) | 0.22 | NA | |
| Cancer | 1 (4.6%) | 1 (5.0%) | 1.11 (0.07–18.90) | 0.95 | 1.05 (0.06–18.40) | 0.97 |
| Diabetes mellitus | 11 (50.0%) | 9 (45.0%) | 0.82 (0.24–2.76) | 0.75 | 0.82 (0.24–2.75) | 0.74 |
| Obesity | 3 (13.6%) | 3 (15.0%) | 1.12 (0.20–6.30) | 0.90 | 1.19 (0.21–6.87) | 0.85 |
| Hypertension | 15 (68.2%) | 12 (60.0%) | 0.70 (0.20–2.49) | 0.58 | 0.72 (0.20–2.57) | 0.61 |
| Chronic heart failure | 1 (4.6%) | 1 (5.0%) | 1.11 (0.07–18.90) | 0.95 | 1.11 (0.07–19.00) | 0.94 |
| Angina/previous MI | 2 (9.1%) | 1 (5.0%) | 0.53 (0.04–6.29) | 0.61 | 0.54 (0.05–6.50) | 0.63 |
| Chronic kidney disease | 3 (13.6%) | 4 (20.0%) | 1.58 (0.31–8.15) | 0.58 | 1.56 (0.30–8.07) | 0.60 |
| Dialysis | 2 (9.1%) | 2 (10.0%) | 1.11 (0.11–8.73) | 0.92 | 1.06 (0.13–8.50) | 0.95 |
| White blood cells, ×10$^9$/mL | 8300 (5145–10248) | 8175 (7200–9925) | 1.00 (1.00–1.00) | 0.52 | NA | |
| Lymphocytes, ×10$^6$/mL | 707 (430–905) | 595 (442–893) | 1.00 (1.00–1.00) | 0.68 | 1.00 (1.00–1.00) | 0.75 |
| Platelet count, ×10$^3$/mL | 161.0 (114.5–205.0) | 182.5 (131.3–221.3) | 1.00 (1.00–1.00) | 0.61 | 1.00 (1.00–1.00) | 0.58 |
| Lactate dehydrogenase, U/L | 438 (323–523) | 378 (333–479) | 1.00 (1.00–1.00) | 0.99 | 1.00 (1.00–1.00) | 0.96 |
| Creatine kinase, U/L | 96 (67–121) | 141 (54–233) | 1.00 (1.00.-1.00) | 0.65 | 1.00 (1.00–1.00) | 0.60 |
| Aspartate transaminase, U/L | 38 (28–51) | 46 (31–65) | 1.01 (0.99–1.03) | 0.44 | 1.01 (0.99–1.03) | 0.45 |
| Alanine transaminase, U/L | 25 (17–38) | 29 (19–42) | 1.01 (0.991.04) | 0.31 | 1.01 (0.99–1.04) | 0.30 |
| Total bilirubin, mg/dL | 0.6 (0.4–0.7) | 0.79 (0.48–1.0) | 5.31 (0.79–35.60) | 0.09 | 5.46 (0.81–36.90) | 0.08 |
| Blood urea nitrogen, mg/dL | 18.0 (12.3–32.0) | 20 (18–34) | 1.00 (0.98–1.03) | 0.51 | 1.01 (0.98–1.04) | 0.48 |
| Serum creatinine, mg/dL | 0.67 (0.53–0.91) | 1.12 (0.85–1.67) | 1.04 (0.90–1.22) | 0.58 | NA | |
| eGFR, mL/min/m$^2$ | 83.2 (63.8–88.6) | 50.1 (34.8–68.7) | 0.98 (0.96–1.00) | 0.04 | NA | |
| C-reactive protein, mg/dL | 8.58 (6.5–12.65) | 11.92 (6.52–16.33) | 1.07 (0.98–1.18) | 0.15 | 1.07 (0.97–1.18) | 0.16 |
| PT-INR | 1.02 (0.99–1.06) | 1.08 (1.02–1.11) | 6.40 (0.08–529.00) | 0.41 | 6.47 (0.07–566.00) | 0.41 |
| APTT, sec | 33.5 (27.9–37.7) | 33.0 (30.9–40.5) | 1.07 (0.98–1.16) | 0.13 | 1.07 (0.99–1.17) | 0.11 |
| Fibrinogen, mg/dL | 561 (477–614) | 484 (437–644) | 1.00 (1.00–1.010 | 0.79 | 1.00 (1.00–1.01) | 0.78 |
| Antithrombin activity, % | 83.2 (76.2–90.0) | 75.7 (70.1–79.6) | 0.94 (0.88–1.00) | 0.03 | 0.94 (0.88–1.00) | 0.03 |
| D-dimer, μg/mL | 1.7 (1.3–2.7) | 1.6 (0.9–2.5) | 1.00 (0.95–1.06) | 0.97 | 1.00 (0.95–1.06) | 0.92 |

Multivariate analyses were performed after adjusting for disease severity at ICU admission based on APACHE II scores. Data are presented as the median (interquartile range) or number (%). NA, not applicable; APACHE II, Acute Physiology and Chronic Health Evaluation II score; SOFA, Sequential Organ Failure Assessment score; MI, myocardial infarction; eGFR, estimated glomerular filtration rate; PT-INR, the international normalized ratio of prothrombin time; APTT, activated partial thromboplastin time.

Table 4 shows the durations of ICU and hospital stays and clinical outcomes in the study subjects. The duration of ICU stay differed between all groups, being longer in the ECMO, MV-only and No MV groups in descending order. The duration of hospital stay was significantly longer in the MV-only and ECMO groups than in the No MV group. However, there was no significant difference between the MV-only and ECMO groups. The percentage of patients discharged home was higher in the No MV group, MV-only group and ECMO group, in that order. The percentage of patients discharged to another acute care hospital was highest in the ECMO group, and the percentage of patients discharged to a long-term care center was highest in the MV-only group. There were no deaths in the No MV group, although four and three patients, respectively, in the MV-only and ECMO groups, died in the ICU or ward.

**Table 4. Patient outcomes.**

| | All (n = 66) | No mechanical ventilation (n = 24) | Mechanical ventilation-only (n = 22) | ECMO (n = 20) |
|---|---|---|---|---|
| Duration of ICU stay * | 11.0 (5.0–31.5) | 4.0 (2.8–5.3) | 14.0 (10.3–29.0) | 36.0 (21.0–46.0) |
| Duration of hospital stay † | 21.0 (15.5–41.0) | 16.0 (13.0–20.3) | 30.0 (20.0–41.5) | 52.0 (32.0–75.0) |
| Discharged to home | 37 (56.1%) | 20 (83.3%) | 11 (50.0%) | 6 (30.0%) |
| Discharged to another acute hospital | 10 (16.7%) | 2 (8.3%) | 1 (4.5%) | 7 (35.0%) |
| Discharged to long-term care center | 9 (13.6%) | 2 (8.3%) | 5 (22.7%) | 2 (10.0%) |
| Still hospitalized | 3 (4.5%) | 0 | 1 (4.5%) | 2 (10.0%) |
| Death | 7 (10.6%) | 0 | 4 (18.2%) | 3 (15.0%) |

Data are presented as the median (interquartile range) or number (%). We used Kruskal-Wallis ANOVA with post-hoc Bonferroni test to compare the durations of ICU and hospital stays. Since the dead were excluded from the ICU and hospital stay duration analysis, the number of patients in the mechanical ventilation-only and ECMO groups was 18 and 17, respectively. * P value was less than 0.05 among all groups. † P value was less than 0.05 except between the mechanical ventilation-only and ECMO groups.

Details about the medications, renal replacement therapy and airway management after ICU admission in each group are shown in S1 Table. A complete dataset of individual data used for analysis is presented in S2 Table.

## Discussion

This study retrospectively investigated risk factors for the need for subsequent MV and ECMO in COVID-19 patients admitted to the ICU. The following items at ICU admission were identified as risk factors for subsequent MV: advanced age, higher APACHE II and SOFA scores, higher C-reactive protein levels, lower lymphocyte count, higher LDH levels, and lower antithrombin levels. Higher LDH levels and BMI were independent risk factors for the need for MV after adjusting for APACHE II score. Further, lower eGFR and lower antithrombin levels were associated with subsequent ECMO requirements. Lower antithrombin levels remained an independent risk factor for ECMO after adjustment for APACHE II score.

Previous studies showed the following risk factors for MV in COVID-19 patients: acute respiratory distress syndrome [8], lower antithrombin levels [17], hypercoagulative state on viscoelastic tests [18], disseminated intravascular coagulopathy (DIC) [19], lower ratio of oxygen saturation (ROX) index [10, 20, 21], older age [22], male sex [22], higher C-reactive protein [23] and LDH [21] levels, lower serum creatinine levels [21], and higher D-dimer levels [24]. Our study results were consistent with these studies.

We demonstrated that higher LDH levels are an independent risk factor for the need for MV. Many earlier studies have shown the association between high LDH levels and COVID-19 severity [2, 8, 25]. Moreover, one other study also showed a correlation between high LDH levels and the need for MV in COVID-19 patients [21], which is consistent with our findings. After adjusting for APACHE II scores, BMI was also identified as an independent risk factor for MV in this study. BMI has also been previously reported as a risk factor for in-hospital mortality and morbidity in patients with COVID-19 [26–28], which is consistent with the results of this study.

Although many cohort studies investigated the characteristics and outcomes of severe COVID-19 patients, few studies have investigated differences in COVID-19 patients with and without ECMO. A previous prospective study that validated a urinalysis-based prediction model of COVID-19 showed that abnormal urinalysis results on admission correlated with

mortality, MV or ECMO requirements [29]. Since chronic kidney disease and pre-existing hemodialysis treatment were reported as risk factors for severe COVID-19 [30], it is plausible that lower eGFR is associated with the need for ECMO in COVID-19 patients.

We found that antithrombin levels were significantly lower in the more severe COVID-19 patients group. Although the correlation between outcomes and antithrombin levels in COVID-19 patients has been reported in several previous studies, those results were contro-versial. Several studies showed that lower antithrombin levels were associated with higher mortality in patients with COVID-19 [17, 29, 31]. Conversely, several reports suggested that antithrombin levels were not associated with severity or mortality in COVID-19 patients [4, 32]. These discrepancies might have been due to the background of the patients studied. Although previous studies recruited COVID-19 patients in various disease stages, our study only included patients admitted to the ICU. Although the results of our study do not reveal the mechanism by which antithrombin levels decrease in severe cases, the following possible mechanisms should be considered. Antithrombin might be consumed by hypercoagulation due to higher inflammation and vascular endothelial damage in severe cases [19, 33, 34]. Alter-natively, increased vascular permeability caused by inflammation in severe cases might result in leakage of antithrombin out of blood vessels [35]. In terms of treatment, the administration of antithrombin concentrates has reduced the mortality rate of patients with severe septic DIC [36, 37]. Although the pathophysiology of DIC associated with COVID-19 differs from that of septic DIC [38, 39], antithrombin supplementation might also improve outcomes in severe COVID-19 patients with DIC [40]. Further investigation is needed to determine the potential therapeutic application of antithrombin in the future.

This study has several limitations. First, this was a retrospective observational study with a limited sample size. Second, since this study enrolled patients with COVID-19 who developed the disease before January 31, 2021, the current risk factors for the subsequent need of MV and ECMO might differ from our findings because the currently prevalent SARS-CoV-2 strains, including BA.5 variants, are different from those at the time the study was conducted. Third, since we could not follow patients who were discharged to another acute hospital or long-care center, their precise outcomes were unclear. Fourth, there were no standard meth-ods or protocols defining the specific criteria or indications for MV and ECMO support; hence, decision-making for the introduction of MV and ECMO differed by hospital and physi-cian. Fifth, we collected laboratory data only once, at the time of ICU admission, and changes in these variables over time were not included in the analyses.

Early recognition of COVID-19 patients with a high risk of exacerbation allows physicians to make adequate clinical decisions, including the need for hospitalization, strict monitoring and medication. Identification of high-risk patients will improve their outcomes, and would enable more appropriate distribution of medical resources.

## Conclusions

We showed that low antithrombin level at ICU admission might be a risk factor for subsequent ECMO requirements, in addition to other previously reported factors.

## Supporting information

**S1 Table. Medications, renal replacement therapy and airway management during ICU admission.** Data are presented as the median (interquartile range) or number (%). Other med-ications included hydroxychloroquine, favipiravir and lopinavir-ritonavir. The total number of patients receiving renal replacement therapy indicates the number of patients who under-went hemodialysis and/or continuous hemodiafiltration. † Mechanical ventilation-only vs.

ECMO group, P < 0.01, Mann-Whitney U test. Patients who died were excluded from the analysis. ‡ Two patients in the ECMO group who were intubated before ICU admission were excluded from the analysis. HD, hemodialysis; CHDF, continuous hemodiafiltration; NA, not applicable.
(DOCX)

**S2 Table. Complete dataset of all the study subjects that was used for analysis.** In the table, presence of the factor in the individual was indicated as '1', while absence of the factor was indicated as '0'. Please see the regular tables for units. Obesity was defined as body mass index of over 30 kg/m$^2$. BMI, body mass index; APACHE II, Acute Physiology and Chronic Health Evaluation II score; SOFA, Sequential Organ Failure Assessment score; CHF, chronic heart failure; OMI, old myocardial infarction; CRF, chronic renal failure; WBC, white blood cells; LDH, lactate dehydrogenase; CK, creatinine kinase; AST, aspartate aminotransferase; ALT; alanine aminotransferase; BUN, blood urea nitrogen; eGFR, estimated glomerular filtration rate; CRP; C-reactive protein; PT-INR, international normalized ratio of prothrombin time; APTT, activated partial thromboplastin time; AT; antithrombin activity; MV, mechanical ventilation; HD, hemodialysis; CHDF; continuous hemodiafiltration; GUH, Gunma University Hospital; JRCMH; Japanese Red Cross Maebashi Hospital; OMH, Ota Memorial Hospital; ND, no data; NA, not applicable.
(XLSX)

## Author Contributions

**Conceptualization:** Ryo Takada, Tomonori Takazawa.

**Data curation:** Ryo Takada, Tomonori Takazawa, Yoshihiko Takahashi, Kenji Fujizuka, Kazuki Akieda.

**Formal analysis:** Ryo Takada, Tomonori Takazawa.

**Investigation:** Ryo Takada, Tomonori Takazawa, Yoshihiko Takahashi, Kenji Fujizuka, Kazuki Akieda.

**Methodology:** Ryo Takada, Tomonori Takazawa.

**Project administration:** Tomonori Takazawa, Shigeru Saito.

**Supervision:** Shigeru Saito.

**Validation:** Ryo Takada, Tomonori Takazawa, Yoshihiko Takahashi, Kenji Fujizuka, Kazuki Akieda, Shigeru Saito.

**Visualization:** Tomonori Takazawa.

**Writing – original draft:** Ryo Takada, Tomonori Takazawa.

**Writing – review & editing:** Ryo Takada, Tomonori Takazawa, Yoshihiko Takahashi, Kenji Fujizuka, Kazuki Akieda, Shigeru Saito.

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
