## [Decision Letter · Decision Letter 0]

27 Sep 2022

PONE-D-22-26070Risk factors for mechanical ventilation and ECMO in COVID-19 patients admitted to ICU: A multi-center retrospective observation studyPLOS ONE

Dear Dr. Takazawa,

Thank you for submitting your manuscript to PLOS ONE. After careful consideration, we feel that it has merit but does not fully meet PLOS ONE’s publication criteria as it currently stands. Therefore, we invite you to submit a revised version of the manuscript that addresses the points raised during the review process.

ACADEMIC EDITOR: Manuscript ID PONE-D-22-26070 entitled "Risk factors for mechanical ventilation and ECMO in COVID-19 patients admitted to ICU: A multi-center retrospective observation study" which you submitted to the PLOS ONE, has been reviewed. A minor revision decision is applied to your manuscript according to the reviewer comments.

We look forward to receiving your revised manuscript.

Kind regards,

Gulali Aktas

Academic Editor

PLOS ONE

Additional Editor Comments :

Dear Dr. Takazawa

Manuscript ID PONE-D-22-26070 entitled "Risk factors for mechanical ventilation and ECMO in COVID-19 patients admitted to ICU: A multi-center retrospective observation study" which you submitted to the PLOS ONE, has been reviewed. The comments of the reviewer(s) are included at the bottom of this letter.

The reviewer(s) have suggest some minor revisions to your manuscript. Therefore, I invite you to respond to the reviewer(s)' comments and revise your manuscript.

Comments of Reviewer 1

Manuscript is a clear, concise, and well-written with relevant introduction. Conclusion in abstract should be written more clearly and precisely. In Methodology explain terms PACHE II score, SOFA score, and how did you calculate them.

Some values in Table 1 and others are express as numbers and percentages, please write n(%) for that variables. If it is possible, please divide Table 2 and Table 3 into two or more tables.

Comments of Reviewer 2

The manuscript PONE-D-22-26070 is a well prepared original article about risk factors of intensive care either with ECMO or mechanical ventilation in Covid-19 patients. I congratulate authors for such a good work. My comments about the sections of the manuscript are as following:

Title is relevant as keywords after abstract which is an adequate summary of the study.

Rationale and aims are clearly mentioned in introduction

Methods and the study design expressed perfectly. Statistical analyses are correct and adequate for the study design.

Presentation of the results is clear. Tables improved the readiness. However, unit of all variables must be stated in the tables. I advise n,% for categorical variables.

Discussion is fair enough. Yet, I suggest discussing the role of inflammatory burden on necessity of ECMO or mechanical ventilation. Studies in literature reported higher inflammatory markers in serious Covid-19 cases compared to the patients with mild or moderate disease (Hematology 2021;26(1):529-542. DOI: 10.1080/16078454.2021.1950898). Discuss.

Conclusions must be clarified. The statement 'This study revealed several risk factors on ICU admission that are related to the 261 need for subsequent MV or ECMO in COVID-19 patients admitted to the ICU.' is not the best conclusion that could be drawn from the study.

Reviewers' comments:

Reviewer's Responses to Questions

**Comments to the Author**

1. Is the manuscript technically sound, and do the data support the conclusions?

Reviewer #1: Yes

Reviewer #2: Yes

2. Has the statistical analysis been performed appropriately and rigorously? 

Reviewer #1: Yes

Reviewer #2: Yes

3. Have the authors made all data underlying the findings in their manuscript fully available?

Reviewer #1: Yes

Reviewer #2: Yes

4. Is the manuscript presented in an intelligible fashion and written in standard English?

Reviewer #1: Yes

Reviewer #2: Yes

5. Review Comments to the Author

Reviewer #1: Manuscript is a clear, concise, and well-written with relevant introduction. Conclusion in abstract should be written more clearly and precisely. In Methodology explain terms PACHE II score, SOFA score, and how did you calculate them.

Some values in Table 1 and others are express as numbers and percentages, please write n(%) for that variables. If it is possible, please divide Table 2 and Table 3 into two or more tables.

Reviewer #2: The manuscript PONE-D-22-26070 is a well prepared original article about risk factors of intensive care either with ECMO or mechanical ventilation in Covid-19 patients. I congratulate authors for such a good work. My comments about the sections of the manuscript are as following:

Title is relevant as keywords after abstract which is an adequate summary of the study.

Rationale and aims are clearly mentioned in introduction

Methods and the study design expressed perfectly. Statistical analyses are correct and adequate for the study design.

Presentation of the results is clear. Tables improved the readiness. However, unit of all variables must be stated in the tables. I advise n,% for categorical variables.

Discussion is fair enough. Yet, I suggest discussing the role of inflammatory burden on necessity of ECMO or mechanical ventilation. Studies in literature reported higher inflammatory markers in serious Covid-19 cases compared to the patients with mild or moderate disease (Hematology 2021;26(1):529-542. DOI: 10.1080/16078454.2021.1950898). Discuss.

Conclusions must be clarified. The statement 'This study revealed several risk factors on ICU admission that are related to the 261 need for subsequent MV or ECMO in COVID-19 patients admitted to the ICU.' is not the best conclusion that could be drawn from the study.

6. PLOS authors have the option to publish the peer review history of their article (what does this mean?). If published, this will include your full peer review and any attached files.

Reviewer #1: No

Reviewer #2: **Yes: **Ozge Kurtkulagi

---

## [Author Response · Author response to Decision Letter 0]

19 Oct 2022

Comments of Reviewer 1

Manuscript is a clear, concise, and well-written with relevant introduction.

Conclusion in abstract should be written more clearly and precisely.

Reply: We have modified the conclusion in the abstract according to your suggestion.

In Methodology explain terms APACHE II score, SOFA score, and how did you calculate them.

Reply: We have added a description of APACHE II and SOFA scores in the Methods section Further, we have cited references for where more information on these scores can be obtained.

Some values in Table 1 and others are express as numbers and percentages, please write n(%) for that variables.

Reply: We have changed the display of the variables in Table 1, as recommended.

If it is possible, please divide Table 2 and Table 3 into two or more tables.

Reply: Tables 2 and 3 are certainly large, but we would like to present the findings at a glance. Hence, we would prefer to retain the table in its current form, if there are no space constraints according to the journal requirements.

Comments of Reviewer 2

The manuscript PONE-D-22-26070 is a well prepared original article about risk factors of intensive care either with ECMO or mechanical ventilation in Covid-19 patients. I congratulate authors for such a good work. My comments about the sections of the manuscript are as following:

Title is relevant as keywords after abstract which is an adequate summary of the study.

Rationale and aims are clearly mentioned in introduction

Methods and the study design expressed perfectly. Statistical analyses are correct and adequate for the study design.

Presentation of the results is clear. Tables improved the readiness. However, unit of all variables must be stated in the tables. I advise n,% for categorical variables.

Reply: We have mentioned the units of the variables in Table 1, as recommended.

Discussion is fair enough. Yet, I suggest discussing the role of inflammatory burden on necessity of ECMO or mechanical ventilation. Studies in literature reported higher inflammatory markers in serious Covid-19 cases compared to the patients with mild or moderate disease (Hematology 2021;26(1):529-542. DOI: 10.1080/16078454.2021.1950898).

Reply: Thank you for presenting this important article. The article emphasizes a decrease in lymphocytes as an indicator of COVID-19 severity, especially when compared to neutrophils and platelets. We have cited this article as Reference 9.

Conclusions must be clarified. The statement 'This study revealed several risk factors on ICU admission that are related to the 261 need for subsequent MV or ECMO in COVID-19 patients admitted to the ICU.' is not the best conclusion that could be drawn from the study.

Reply: We have modified the conclusion according to your suggestion.

Reply: We have made revisions to ensure that the manuscript matches the style of PLOS ONE.

Reply: We have added the following sentence as additional details regarding participant consent: Since the ethics committees of the participating institutions waived the need for obtaining verbal or written consent from the patients participating in this study, patient informed consent was obtained in the form of the option to opt-out on the institutions’ websites.

The current study did not include minors.

Reply: In this revision, we provide the minimal underlying dataset of the study as Supplementary Table S2. The table contains anonymized patient data.

Reply: We confirmed that the title on the online submission form (via Edit Submission) and the title in the manuscript are identical.

Reply: We have removed the ethics statement apart from that in the methods section.

Reply: We have confirmed that the reference list is complete and correct. The references we have cited do not include retracted articles.

---

## [Decision Letter · Decision Letter 1]

2 Nov 2022

Risk factors for mechanical ventilation and ECMO in COVID-19 patients admitted to the ICU: A multicenter retrospective observational study

PONE-D-22-26070R1

Dear Dr. Takazawa,

We’re pleased to inform you that your manuscript has been judged scientifically suitable for publication and will be formally accepted for publication once it meets all outstanding technical requirements.

Kind regards,

Gulali Aktas

Academic Editor

PLOS ONE

Additional Editor Comments (optional):

Authors were very responsive to the reviewers' comments. The manuscript is improved significantly. Therefore it is acceptable for publication.

Reviewers' comments:

Reviewer's Responses to Questions

**Comments to the Author**

1. If the authors have adequately addressed your comments raised in a previous round of review and you feel that this manuscript is now acceptable for publication, you may indicate that here to bypass the “Comments to the Author” section, enter your conflict of interest statement in the “Confidential to Editor” section, and submit your "Accept" recommendation.

Reviewer #1: All comments have been addressed

Reviewer #2: All comments have been addressed

2. Is the manuscript technically sound, and do the data support the conclusions?

Reviewer #1: Yes

Reviewer #2: Yes

3. Has the statistical analysis been performed appropriately and rigorously? 

Reviewer #1: Yes

Reviewer #2: Yes

4. Have the authors made all data underlying the findings in their manuscript fully available?

Reviewer #1: Yes

Reviewer #2: Yes

5. Is the manuscript presented in an intelligible fashion and written in standard English?

Reviewer #1: Yes

Reviewer #2: Yes

6. Review Comments to the Author

Reviewer #1: (No Response)

Reviewer #2: All review suggestions were followed b y the authors. Well done. There is nothing more that require further revision. I recommend publication.

7. PLOS authors have the option to publish the peer review history of their article (what does this mean?). If published, this will include your full peer review and any attached files.

Reviewer #1: No

Reviewer #2: No

---

## [Editor Report · Acceptance letter]

4 Nov 2022

PONE-D-22-26070R1 

Risk factors for mechanical ventilation and ECMO in COVID-19 patients admitted to the ICU: A multicenter retrospective observational study 

Dear Dr. Takazawa:

I'm pleased to inform you that your manuscript has been deemed suitable for publication in PLOS ONE. Congratulations! Your manuscript is now with our production department. 

Kind regards, 

on behalf of

Professor Gulali Aktas 

Academic Editor

PLOS ONE